# Promiscuous Lipase-Catalyzed Knoevenagel–Phospha–Michael Reaction for the Synthesis of Antimicrobial β-Phosphono Malonates

**DOI:** 10.3390/ijms23158819

**Published:** 2022-08-08

**Authors:** Jan Samsonowicz-Górski, Dominik Koszelewski, Paweł Kowalczyk, Paweł Śmigielski, Anastasiia Hrunyk, Karol Kramkowski, Aleksandra Wypych, Mateusz Szymczak, Rafał Lizut, Ryszard Ostaszewski

**Affiliations:** 1Institute of Organic Chemistry, Polish Academy of Sciences, 01-224 Warsaw, Poland; 2Department of Animal Nutrition, The Kielanowski Institute of Animal Physiology and Nutrition, Polish Academy of Sciences, 05-110 Jabłonna, Poland; 3Department of Physical Chemistry, Medical University of Bialystok, 15-089 Białystok, Poland; 4Centre for Modern Interdisciplinary Technologies, Nicolaus Copernicus University, 87-100 Toruń, Poland; 5Department of Molecular Virology, Institute of Microbiology, Faculty of Biology, University of Warsaw, 02-096 Warsaw, Poland; 6Institute of Mathematics, Informatics and Landscape Architecture, The John Paul II Catholic University of Lublin, 20-708 Lublin, Poland

**Keywords:** enzyme promiscuity, the phospha-Michael addition, antimicrobial activity, lipase, *Candida cylindracea*, bio catalysis, β-phosphonomalononitriles, *E. coli* cells

## Abstract

An enzymatic route for phosphorous–carbon bond formation was developed by discovering new promiscuous activity of lipase. We reported a new metal-free biocatalytic method for the synthesis of pharmacologically relevant β-phosphonomalononitriles via a lipase-catalyzed one-pot Knoevenagel–phospha–Michael reaction. We carefully analyzed the best conditions for the given reaction: the type of enzyme, temperature, and type of solvent. A series of target compounds was synthesized, with yields ranging from 43% to 93% by enzymatic reaction with *Candida cylindracea* (CcL) lipase as recyclable and, a few times, reusable catalyst. The advantages of this protocol are excellent yields, mild reaction conditions, low costs, and sustainability. The applicability of the same catalyst in the synthesis of β-phosphononitriles is also described. Further, the obtained compounds were validated as new potential antimicrobial agents with characteristic *E. coli* bacterial strains. The pivotal role of such a group of phosphonate derivatives on inhibitory activity against selected pathogenic *E. coli* strains was revealed. The observed results are especially important in the case of the increasing resistance of bacteria to various drugs and antibiotics. The impact of the β-phosphono malonate chemical structure on antimicrobial activity was demonstrated. The crucial role of the substituents attached to the aromatic ring on the inhibitory action against selected pathogenic *E. coli* strains was revealed. Among tested compounds, four β-phosphonate derivatives showed an antimicrobial activity profile similar to that obtained with currently used antibiotics such as ciprofloxacin, bleomycin, and cloxacillin. In addition, the obtained compounds constitute a convenient platform for further chemical functionalization, allowing for a convenient change in their biological activity profile. It should also be noted that the cost of the compounds obtained is low, which may be an attractive alternative to the currently used antimicrobial agents. The observed results are especially important because of the increasing resistance of bacteria to various drugs and antibiotics.

## 1. Introduction

The struggle with infections caused by antibiotic-resistant pathogens poses a significant threat to global health [1,2]. It has been estimated that by 2050, 10 million worldwide deaths could result from antibiotic resistance, making it deadlier than cancer and resulting in a global economic output cost of USD 100 trillion [3]. Consequently, tackling antimicrobial resistance and searching for new antimicrobial agents will always remain a challenge and a reason for intensive work by medicinal chemists.

Phosphonates are important structural building blocks of several natural products and are also considered one of the most commonly encountered compounds in medicinal chemistry. Phosphonate derivatives possess diverse chemotherapeutic activities, which include enzyme inhibition [4], peptide mimics [5], and anti-microbial [6] and anti-fungal activities [7]. Similarly, nitrile-containing medical agents have emerged as the number of anti-microbial pharmaceuticals has increased [8,9,10] (Figure 1). The prevalence of nitrile-containing pharmaceuticals and the continued stream of potential agents in the clinic attest to the biocompatibility of the nitrile functionality. Due to the large amount of antimicrobial resistance, there is a growing need to elaborate lead structural scaffolds that may be useful in developing potent antimicrobial drugs. In the light of the benefits resulting from the antimicrobial activity of these two groups of compounds, it seems logical to search for new compounds combining the unique molecular identity of phosphonates and nitriles in their structures. Therefore, the development of new methods for their preparation is of current concern. Only a few reports have appeared regarding the antimicrobial activity of β-phosphonomalononitriles and β-phosphononitriles, and these are strictly limited to chromen and quinoline moieties (Figure 1) [11,12,13].

Revising the toxic effect of the novel organic compounds, containing both a phosphorus–carbon bond and nitrile function, on bacterial cells can provide suitable antimicrobial agents against bacterial clinical pathogens [14,15,16,17]. A careful analysis of the literature data showed that there are no data related to the influence of the size and electronic effects of aryl and heteroaryl groups in the beta position of the β-phosphonomalononitrile derivatives on their antimicrobial activity (Figure 1). Unfortunately, there are no mild and environmentally sustainable methods to synthesize these types of compounds. The goal of the presented studies is the development of the efficient enzymatic preparation of β-phosphonomalononitrile derivatives with aryl and heteroaryl groups in the β-position and their validation and comparison with commonly used antimicrobial agents against model strains of Escherichia coli K12 (with native LPS in its structure) and R2–R4 (LPS of different lengths in its structure).

The phospha–Michael reaction of phosphite nucleophile to carbon–carbon double bonds is a widely used method for carbon–phosphorus bond formation [18,19,20]. The reaction is usually catalyzed by alkaline earth metal oxides [21], functionalized silica [22], organic bases [23,24], or iron-doped carbon nanotubes [25]. Many of these two-component protocols essentially involve the addition of phosphite nucleophiles to benzylidine malononitrile; however, reports on the multicomponent one-pot synthesis of β-phosphonomalononitriles which eliminate the need for α,β-unsaturated malonates derivatives are highly limited [26,27,28]. Unfortunately, many of the reported protocols suffer from some inconveniences, such as the stoichiometric amount of catalysts, water free environment, or the application of expensive and highly toxic catalysts [29,30]. Moreover, some of the available methods lead to obtaining products contaminated with metals, which significantly reduces their applicability in the synthesis of biologically active compounds. Thus, it seems meaningful to find a method to overcome these limitations.

## 2. Results and Discussion

### 2.1. Chemistry

Among hydrolases, lipases (EC 3.1.1.3) [31,32] have appeared as a leading class of biocatalysts in organic synthesis. Generally, lipases are employed to perform three types of reactions: hydrolysis, esterification, and transesterification [33]. An increasing number of reports on lipase-catalyzed, unconventional reactions have directed attention towards lipase promiscuity [34,35,36,37,38]. As a continuation of our research on seeking new catalytic activities for hydrolases [39,40,41,42,43], we focused our efforts to elaborate sustainable metal-free methods towards desired β-phosphonomalononitriles and β-phosphononitriles (Figure 2 and Figure 3). Therefore, the possibility of using the biocatalytic approach in the synthesis of target β-phosphonates should be considered. The use of hydrolases as catalysts in various variants of the Michael addition of different nucleophilic partners has been widely discussed in the literature [44,45,46]. Recently, it was also shown that lipases catalyze aldol and Knoevenagel condensations [47]. This observation prompted us to investigate the viability of β-phosphonomalononitriles synthesis using the Knoevenagel–phospha–Michael addition of alkyl phosphites to simultaneously generated α,β-unsaturated malonates in the presence of enzymes as an efficient and sustainable catalyst of both transformations.

Regarding the promiscuous activity of lipases [39], the model multi-component (MCR) Knoevenagel–phospha–Michael reaction of benzaldehyde (1 mmol), malononitrile (1 mmol), and dimethyl phosphite (1 mmol) was conducted in *tert*-butyl methyl ether (TBME) at 20 °C (Figure 1 and Table 1, entry 1). Selected commercially available lipases—one domestically prepared wheat germ lipase and five liver acetone powders—were tested as catalysts, and the results are summarized in Table 1.

As shown in Table 1, lipase from *Candida cylindracea* (CcL) was found as the best catalyst among the tested lipases for this addition reaction (Table 1, entry 2). The β-phosphonomalononitrile **1** was obtained in good yield (75%) after 8 h in neat at 20 °C. The yield did not increase noticeably after 8 h. In the absence of enzyme, only traces of target product **1** was formed (Table 1, entry 1). To confirm the promiscuous activity of CcL in the studied reaction, bovine serum albumin (BSA) [48] and thermally deactivated CcL were also applied. The results indicated that application of BSA provided target product **1** with an 8% yield, while denatured CcL gave only tracers of product **1** (Table 1, entries 25 and 26). Catalytic proficiency in the studied reaction of the BSA can be explained by the basic character of what the attractive feature of this protein is [49]. These results clearly show that the peculiar active site of CcL is responsible for the studied multicomponent reaction. In addition, two different non-enzymatic catalysts reported in the literature as sustainable promoters of Knoevenagel and Michael additions [50,51,52,53], copper (II) acetate and zinc oxide/palladium (II) acetate, were tested under similar reaction conditions leading to target product **1,** with up to 23% yield (Table 1, entries 27–29). It is well recognized that the reaction medium has a great impact on enzyme properties [54]. The influence of several organic solvents on the model reaction was revised (Table 1, entries 2–11), and product formation was observed in all solvents used. However, the initially selected TBME provided the target product **1** with the highest yield (Table 1, entry 2), therefore this solvent was applied in the further optimization. It is worth noting that both water and solvent-free conditions favor the formation of the target product **1** with 42% and 63% yields, respectively (Table 1, entries 10 and 11). This fits perfectly with the assumptions of green chemistry related to the reduction of toxic reagents and solvents [55]. Next, the model reaction was carried out at elevated temperatures. The reaction yield increased to 83% at 30 °C and decreased above 30 °C, which may be due to changes in the quaternary structure of the enzyme used (Table 1, entries 12 and 13). Further, the yield of target compound **1** increased negligibly (85%, Table 1, entry 14) by raising the amount of CcL from 50 mg to 80 mg Thus, 50 mg of CcL was found as the optimal amount for the further investigations. Unfortunately, the obtained chiral product remains in agreement with our previous observations for lipase’s promiscuous activity [39].

The reusability of an enzyme is an important factor that significantly reduces overall costs of the method. In this work, lipase from *Candida cylindracea* was reused up to five times with a gradual decrease in the yield, up to 43% after the fourth cycle and 32% after the fifth one.

Finally, the elaborated enzymatic protocol enabled for synthesis of series of β-phosphonomalononitriles and β-phosphononitriles **2**–**12** with good to very high yields for aromatic and aromatic aldehydes (Figure 3). The enzymatic phospha–Michael addition with aliphatic aldehyde provided product **10** with lower yield of 43% (Figure 3). Similar reductions in reaction yields were observed for sterically bulky electron rich aldehydes with two methoxy groups located at the phenyl ring, which resulted in product 4, with a 64% yield. As shown in Figure 3, the application of ethyl 2-cyanoacetate as the substrate provided products **11** and **12,** with yields up to 89% as a mixture of diasteroisomers (ratio: 2:1). 

Additional experiments were performed to obtain insights on reaction pathways. Under developed conditions, benzylidenemalononitrile was used together with dimethyl *H*-phosphite in the presence of CcL as a catalyst which resulted in target β-phosphonomalononitriles **1** in a high yield of 88%. This observation is supported by the literature reports regarding the Knoevenagel reaction promoted by lipases [42,56] constituting the initial formation of an α,β-unsaturated intermediate in the presence of CcL (Step 1, Figure 2).

To our delight, the developed conditions of the enzymatic addition reaction allowed for the phospha–Michael addition of two different *H*-phosphonates to acrylonitrile, which resulted in dibenzyl (2-cyanoethyl)phosphonate (**13**) and diethyl (2-cyanoethyl)phosphonate (**14**), with 43% and 49% yields, respectively (Figure 3). Compound **14** is an intermediate for the preparation of triose-phosphate isomerase inhibitors [57]. This opens up new possibilities for the use of lipase promiscuity in the formation of phosphorus–carbon bonds. The structures of new compounds were confirmed using NMR and mass spectroscopy. Spectral data of known compounds remained in agreement with the literature data. The NMR spectra of compounds **1**–**14** are presented in the experimental section (Appendix A). 

### 2.2. Cytotoxic Studies of the Library of β-Phosphonate Derivatives ***1***–***14***

The toxic effect on bacterial cells after the analysis of the MIC and MBC tests for all 14 analyzed compounds, for which the MIC values were observed in the range of 0.2–1.4 µg/mL, and 2–82 µg/mL for MBC values in the analyzed model strains K12, R2, R3 and R4) (Figure 4 and Figure 5), which had specific functional groups in the structure of the aromatic ring located at the beta position. It is worth noting that the introduction of a halogen atom into the structure of the tested compounds had a significant effect on the activity of compinds 6 and 7, which is often observed for various types of compounds exhibiting antibacterial activity [58,59,60]. Similar enhanced antimicrobial activity was observed for methyl and methoxy groups located in the aromatic ring of the studied compounds **5** and **12** (Figure 4, Figure 5 and Figure 6). The pivotal role of methyl groups on antibacterial activity was reported for some heteroaromatic agents [61,62].

The analyzed bacterial strains used in the experiments were used in 48-well plates. (Figure 4, Figure 5 and Figure 6 and Table 2).

### 2.3. Analysis of R2–R4 E. coli Strains Modified with β-Phosphonate Derivatives

The obtained MIC values as well as our previous studies with various types of the analyzed compounds [63,64,65,66] indicate that β-phosphonate derivatives also show a strong toxic effect on the analyzed bacterial model strains. The three analyzed compounds were selected for further analysis by modifying *E. coli*’s DNA. Modified bacterial DNA was digested with Fpg as described earlier [67,68,69,70]. All selected analyzed β-phosphonate derivatives (Figure 3) with different substituents located at the phenyl ring are responsible for the change of the topology of bacterial DNA. After digestion with Fpg, approximately 3.5% of oxidative damage was identified, which very strongly indicates oxidative damage in bacterial DNA, similar to the previous observations [71,72,73,74]. 

The obtained results indicate that all tested β-phosphono malonates show cytotoxic activity in all analyzed *E. coli* strains differing in LPS length. Different inhibitory activity was found depending on the nature of the R1 and R2 substituents attached to the phosphorus–carbon bond and nitrile function of the tested compounds. Among all tested compounds, the compounds from **5**–**7** and **12** showed a stronger antibacterial effect than the others. It is worth noting that the introduction of the phosphorus–carbon bond and nitrile function into the structure of the tested compounds had a significant impact on their activity and cytotoxicity and high selectivity against selected *E. coli* model strains in the MIC and MBC tests, which is often observed in various types of compounds showing strong microbiological activity on cells [63,64,65,66]. These compounds showed higher activity against strains R2, R3, and R4 than commonly used antibiotics (Figure 4, Figure 5, Figure 6 and Figure 7). The values of the MIC and MBC tests for each model of *E. coli* R2–R4 and K12 strains were visible on all analyzed growth microplates after the addition of resazurin. The analyzed bacterial strains used in the experiments were used in 48-well plates which were treated with the analyzed compounds in the MIC and MBC assays. On the basis of their analysis, color changes were observed for all tested compounds but at different levels and at different dilutions. The most sensitive to the effects of the analyzed compounds were the bacterial strains R3 and R4 due to the increasing length of their LPS (visible dilutions 10−2 corresponding to a concentration of 0.0015 μM) more than strains K12 and R2 (visible dilutions of 10−6 corresponding to a concentration of 0.0015 μM). Strain R4 was the most sensitive, possibly due to the longest length of lipopolysaccharide (LPS) in the bacterial membrane. In all analyzed cases, the MBC test values were approximately 80 times higher than the MIC test values in eight analyzed compounds, including those in Figure 4, Figure 5 and Figure 6 and Table 2.

### 2.4. R2–R4 E. coli Strains with Tested β-Phosphonate Derivatives

The performed studies prove that the analyzed compounds can potentially be used as “substitutes” for the currently used antibiotics in hospital and clinical infections (Figure 7, Figure 8, Figure 9 and Appendix A).

Large modifications of plasmid DNA were observed for the analyzed compounds **5**, **6**, **7** and **12** showing their high superselectivity.

## 3. Materials and Methods 

### 3.1. Microorganisms and Media

*E. coli* K-12, R1–R4 strains were received from Prof. Jolanta Łukasiewicz at the Ludwik Hirszfeld Institute of Immunology and Experimental Therapy (Polish Academy of Sciences, Warsaw, Poland). Bacteria were cultivated in a tryptic soy broth (TSB; Sigma-Aldrich, Saint Louis, MO, USA) liquid medium and on agar plates containing TSB medium. *N*,*N*-Dimethylformamide (DMF) was purchased from Sigma Aldrich (CAS No. 68-12-2, Poznań, Poland). Lanes 1kb-ladder and Quick Extend DNA ladder (New England Biolabs, Ipswich, MA, USA), with MIC and MBC tests, are described in detail in the previous work [73,74,75,76] and analyzed by the Tukey test indicated by (*p* < 0.05): * *p* < 0.05, ** *p* < 0.1, *** *p* < 0.01. 

### 3.2. Chemicals

All reagents and the solvents were purchased from Sigma-Aldrich. All solvents were of analytical grade and were used without prior distillation. All specific strains such as *Pseudomonas fluorescens* (PfL) (catalogue number 534730, Lot. number MKBH1198V), *Candida rugosa* (CrL) (catalogue number 90860, Lot. number BCBH7102V), *Candida cylindracea* (CcL) (catalogue number 62316, Lot. number 1336707), and bovine serum albumin were purchased from Sigma-Aldrich. Immobilized lipase from *Candida antarctica* B (Novozym 435) (catalogue number LC200223) was purchased from Novo Nordisk. Lipase from porcine pancreas, Type II (PpL) (catalogue number L-3126, Lot. number 108H1379) was purchased from Sigma-Aldrich. Goose, chicken, bovine, wild hog and deer livers were converted to the acetone powder (GLAP) by the method of Connors et al. [77]. Homemade lipase from wheat germ was prepared according to the literature protocol [78]. Merck silica gel plates 60 F254 were used for TLC (Thin Layer Chromatography) analysis. Crude reaction mixtures were purified using column chromatography on Merck silica gel 60/230–400 mesh, with an appropriate mixture of hexane and ethyl acetate as solvent. Nuclear magnetic resonance spectra (NMR) were performed on Varian apparatus (Varian, Saint Louis, MI, USA) (400 MHz) and (500 MHz), mass spectrometer was from Waters Company, Milford, USA. Chemical shifts are expressed in ppm and coupling constant (*J*) in Hz using TMS as an internal standard. High-resolution mass spectra were acquired on a Maldi SYNAPT G2-S HDMS (Waters) apparatus with a QqTOF analyzer. Enzymatic reactions were performed in a vortex (Heidolph Promax 1020) equipped with incubator (Heidolph Inkubator 1000). To prove the ability of the established protocol, each reaction was repeated at least three times. 

### 3.3. General Procedure for the Synthesis of β-Phosphonate Derivatives (***1***–***12***)

A mixture of an aldehyde (1 mmol), *Candida cylindracea* lipase (CcL) (50 mg), malononitrile or ethyl cyanoacetate (1 mmol), and dimethyl phosphite (1 mmol) in TBME (2 mL) was shaken at 200 rpm at 30 °C for 8 h. Reaction was terminated by filtering off the catalyst through the filter funnel with frit. The yields of MCR toward compounds **1**–**12**, catalyzed by *Candida cylindracea* lipase (CcL) are provided in Figure 2. Melting points and spectral data remained in agreement with the literature data for known compounds. The structure of obtained products was confirmed using NMR and mass spectroscopy.

### 3.4. General Procedure for the Synthesis of β-Phosphonate Derivatives (***13***,***14***)

A mixture of an acrylonitrile (1 mmol), *Candida cylindracea* lipase (CcL) (50 mg), and diethyl phosphite or dibenzyl phosphite (1 mmol) in TBME (2 mL) was shaken at 200 rpm at 30 °C for 8 h. Reaction was terminated by filtering off the catalyst through the filter funnel with frit. The yields of compounds **13** and **14**, catalyzed by *Candida cylindracea* lipase (CcL), are provided in Figure 2. 

The obtained MIC values, as well as our previous studies with various types of the analysed compounds [63,64,65,66], indicate that derivatives of β-phosphonate derivatives also show a strong toxic effect of the analyzed model strains of bacteria. The three compounds analyzed were selected for further analysis by modifying their DNA. Modified bacterial DNA was digested with Fpg as previously described [70,71]. All selected analyzed derivatives of β-phosphonate derivatives, including various types of alkoxy groups, substituents located at aromatic rings, and the length of the alkyl chain, can strongly change the bacterial DNA topology. After Fpg digestion, approximately 3.5% of the oxidative damage was identified, which, similar to previous observations, indicates very strong oxidative damage in bacterial DNA [61,62,63,64,65,66,67,68,69,70,71,72,73,74]. Different types of alkoxy groups, substituents located on the aromatic ring and the length of the alkyl chain, may determine the toxicity of the analyzed *E. coli* strains, including, in particular, R4, as evidenced by the obtained MIC, MBC, and MTT values. The obtained results for individual compounds were statistically significant at the level of *p* < 0.05 (Figure 4, Figure 5, Figure 6, Figure 7, Figure 8 and Figure 9).

**Dimethyl (2,2-dicyano-1-phenylethyl)phosphonate (1).** Compound **1** was obtained according to General Method with 83% yield (219 mg, 0.83 mmol) as white solid with m.p. 126–127 °C [Lit. m.p. 126–127 °C; [24]; ^1^H NMR (400 MHz, CDCl_3_) δ 7.52–7.31 (m, 5H), 4.66–4.50 (m, 1H), 3.78 (d, *J* = 11.1 Hz, 3H), 3.72–3.60 (m, 1H), 3.51 (d, *J* = 10.8 Hz, 3H); ^13^C NMR (100 MHz, CDCl_3_) δ 130.28, 130.17, 129.94, 129.89, 129.78, 129.75, 129.55, 129.42, 111.68, 111.56, 111.48, 111.30, 54.95, 54.81, 53.83, 53.68, 45.99, 43.11, 25.70; ^31^P NMR (162 MHz, CDCl_3_) δ 21.8. NMR data were in accordance with those reported in the literature [24].

**Dimethyl (2,2-dicyano-1-(4-nitrophenyl)ethyl)phosphonate (2).** Compound **2** was obtained according to General Method with 70% yield (216 mg, 0.7 mmol) as light yellow solid with m.p. 133–135 °C; ^1^H NMR (400 MHz, CDCl_3_) δ 8.27 (d, *J* = 8.2 Hz, 2H), 7.70 (d, *J* = 6.9 Hz, 2H), 4.72 (dd, *J* = 9.1, 7.4 Hz, 1H), 3.81 (d, *J* = 11.2 Hz, 3H), 3.74 (d, *J* = 11.9 Hz, 1H), 3.65 (d, *J* = 11.0 Hz, 3H); ^13^C NMR (100 MHz, CDCl_3_) δ 148.54, 148.51, 137.48, 137.43, 130.62, 130.56, 124.50, 124.48, 111.07, 110.96, 110.86, 54.75, 54.68, 54.03, 53.96, 52.08, 44.48, 43.05, 25.09; ^31^P NMR (162 MHz, CDCl_3_) δ 20.3. HRMS (ESI) *m*/*z* calcd for C_12_H_13_N_3_O_5_P [M + H] + 310.0587, found 310.0584.

**Dimethyl (2,2-dicyano-1-(4-methoxyphenyl)ethyl)phosphonate (3).** Compound **3** was obtained according to General Method with 91% yield (267 mg, 0.91 mmol) as colorless solid with m.p. 96–97 °C [Lit. m.p. 95–96 °C; [27]; ^1^H NMR (400 MHz, CDCl_3_) δ 7.38 (d, *J* = 8.0 Hz, 2H), 6.94 (d, *J* = 8.0 Hz, 2H), 4.47 (t, *J* = 8.2 Hz, 1H), 3.81 (s, 3H), 3.68 (d, *J* = 11.4 Hz, 3H), 3.59 (dd, *J* = 8.0 Hz, *J* = 18.4 Hz, 1H), 3.54 (d, *J* = 10.8 Hz, 3H); ^13^C NMR (100 MHz, CDCl3) δ 160.5, 130.5, 121.4, 114.9, 111.2 (d, *J* = 9.8 Hz), 111.0 (d, *J* = 12.7 Hz), 55.2, 54.6 (d, *J* = 6.9 Hz), 53.4 (d, *J* = 7.3Hz), 43.8 (d, *J* = 145.4 Hz), 25.7; ^31^P NMR (162 MHz, CDCl_3_) δ 22.10. NMR data were in accordance with those reported in the literature [27].

**Dimethyl (2,2-dicyano-1-(2,4-dimethoxyphenyl)ethyl)phosphonate (4).** Compound **4** was obtained according to General Method with 64% yield (207 mg, 0.64 mmol) as colorless solid with m.p. 114–116 °C; ^1^H NMR (500 MHz, CDCl_3_) δ 7.47 (dd, *J* = 8.6, 2.0 Hz, 1H), 6.56–6.46 (m, 2H), 4.49 (dd, *J* = 10.0, 8.3 Hz, 1H), 4.28 (dd, *J* = 21.5, 8.4 Hz, 1H), 3.84 (s, 3H), 3.80 (s, 3H), 3.79 (d, *J* = 10.8 Hz, 3H), 3.58 (d, *J* = 10.8 Hz, 3H); ^13^C NMR (100 MHz, CDCl_3_) δ 160.5, 130.5, 111.2 (d, *J* = 9.8 Hz), 110.0 (d, *J* = 12.7 Hz), 105.0 (d, *J* = 10.5 Hz), 99.1 (d, *J* = 10.5 Hz), 55.9, 55.4, 54.2 (d, *J* = 6.9 Hz), 53.2 (d, *J* = 7.3Hz), 36.6, 35.4, 25.0; ^31^P NMR (202 MHz, CDCl_3_) δ 23.1. HRMS (ESI) *m*/*z* calcd for C_14_H_18_N_2_O_5_P [M + H] + 325.0947, found 325.0943.

**Dimethyl (2,2-dicyano-1-(4-methylphenyl)ethyl)phosphonate (5).** Compound **5** was obtained according to General Method with 93% yield (254 mg, 0.93 mmol) as colorless solid with m.p. = 109–110 °C [Lit. m.p. 108–110 °C; [27]; ^1^H NMR (400 MHz, CDCl_3_) δ 7.33 (d, *J* = 2.0 Hz, 2H), 7.21 (d, *J* = 8.2 Hz, 2H), 4.57–4.50 (m, 1H), 3.78 (d, *J* = 11.1 Hz, 3H), 3.61 (dd, *J* = 21.3, 8.0 Hz, 1H), 3.53 (d, *J* = 10.8 Hz, 3H), 2.34 (s, 3H); ^13^C NMR (100 MHz, CDCl_3_) δ 139.7, 130.2, 129.0, 126.6, 111.1 (d, *J* = 9.6 Hz), 110.9 (d, *J* = 6.2 Hz), 54.6 (d, *J* = 6.8 Hz), 53.4 (d, *J* = 7.3 Hz), 44.2 (d, *J* = 144.6 Hz), 25.5, 21.1; ^31^P NMR (162 MHz, CDCl3) δ 22.0. NMR data were in accordance with those reported in the literature [28].

**Dimethyl (2,2-dicyano-1-(4-fluorophenyl)ethyl)phosphonate (6).** Compound **6** was obtained according to General Method with 85% yield (240 mg, 0.85 mmol) as colorless solid with m.p. = 97–98 °C; ^1^H NMR (400 MHz, CDCl_3_) δ 7.53–7.44 (m, 2H), 7.19–7.08 (m, 2H), 4.54–4.43 (m, 1H), 3.81 (d, *J* = 11.1 Hz, 3H), 3.64-3.63 (m, 1H), 3.60 (d, *J* = 10.8 Hz, 3H); ^13^C NMR (100 MHz, CDCl_3_) δ 131.23, 116.81, 116.61, 111.01, 110.93, 54.60, 53.63, 53.55, 44.45, 43.00, 25.63; ^31^P NMR (162 MHz, CDCl_3_) δ 21.5. HRMS (ESI) *m*/*z* calcd for C_13_H_15_FN_2_O_4_P [M + H] + 313.0747, found 313.0746.

**Dimethyl (2,2-dicyano-1-(4-chlorophenyl)ethyl)phosphonate (7).** Compound **7** was obtained according to General Method with 76% yield (226 mg, 0.76 mmol) as colorless solid with m.p. 117–118 °C [Lit. m.p. 118 °C; [22]; ^1^H NMR (400 MHz, CDCl_3_) δ 7.42 (d, *J* = 1.2 Hz, 4H), 4.54 (dd, *J* = 9.0, 7.6 Hz, 1H), 3.80 (d, *J* = 11.1 Hz, 3H), 3.66 (d, *J* = 7.6 Hz, 1H), 3.60 (d, *J* = 10.9 Hz, 3H); ^13^C NMR (100 MHz, CDCl_3_) δ 135.99, 135.96, 130.69, 129.81, 129.79, 128.57, 111.15, 111.04, 110.99, 110.87, 54.70, 54.63, 53.69, 53.62, 44.51, 43.07, 25.43; 31P NMR (162 MHz, CDCl3) δ 21.2. NMR data were in accordance with those reported in the literature [22].

**Dimethyl (2,2-dicyano-1-(furan-2-yl)ethyl)phosphonate (8).** Compound **8** was obtained according to General Method with 86% yield (218 mg, 0. 86 mmol) as colorless solid with m.p. 44–45 °C [Lit. m.p. 42 °C; [66]; ^1^H NMR (400 MHz, CDCl_3_) δ 7.50 (d, *J* = 0.8 Hz, 1H), 6.62 (s, 1H), 6.52–6.42 (m, 1H), 4.56–4.46 (m, 1H), 3.98–3.86 (m, 1H), 3.81 (s, 3H), 3.74 (d, *J* = 10.9 Hz, 3H); ^13^C NMR (100 MHz, CDCl_3_) δ 144.2, 111.9, 111.4, 111.3, 110.5, 54.4 (d, *J* = 7.1 Hz), 54.0 (d, *J* = 6.9 Hz), 38.9 (d, *J* = 147.3 Hz), 24.2; ^31^P NMR (162 MHz, CDCl_3_) δ 18.7. NMR data were in accordance with those reported in the literature [27,79].

**Dimethyl (2,2-dicyano-1-(thiophen-2-yl)ethyl)phosphonate (9).** Compound **9** was obtained according to General Method with 69% yield (186 mg, 0.69 mmol) as colorless solid with m.p. 52–53 °C; ^1^H NMR (400 MHz, CDCl_3_) δ 7.52–7.32 (m, 2H), 7.07 (d, *J* = 3.6 Hz, 1H), 4.68–4.50 (m, 1H), 3.99 (dd, *J* = 22.2, 6.5 Hz, 1H), 3.80 (d, *J* = 11.1 Hz, 3H), 3.68 (d, *J* = 11.2 Hz, 3H); ^13^C NMR (100 MHz, CDCl_3_) δ 129.65, 129.57, 127.82, 127.80, 127.63, 127.60, 110.84, 54.70, 53.84, 40.48, 39.00, 36.28, 26.64; ^31^P NMR (162 MHz, CDCl_3_) δ 19.9; HRMS (ESI) *m*/*z* calcd for C_10_H_12_N_2_O_3_PS [M + H] + 271.0301, found 271.0300.

**Dimethyl (2,2-dicyano-1-monooctyl)phosphonate (10).** Compound **10** was obtained according to General Method with 43% yield (129 mg, 0.43 mmol) as colorless semi-solid with ^1^H NMR (400 MHz, CDCl_3_) δ 4.30 (dd, *J* = 13.1, 3.6 Hz, 1H), 3.86 (d, *J* = 5.0 Hz, 3H), 3.83 (d, *J* = 5.1 Hz, 3H), 2.39 (dtd, *J* = 20.0, 7.1, 3.6 Hz, 1H), 2.02–1.77 (m, 2H), 1.55 (p, *J* = 7.9, 7.4 Hz, 2H), 1.34–1.26 (m, 10H), 0.87 (t, *J* = 7.2 Hz, 3H); 13C NMR (100 MHz, CDCl_3_) δ 53.57, 49.97, 45.67, 37.16, 31.73, 29.21, 29.07, 27.17, 23.67, 22.59, 14.03; 31P NMR (162 MHz, CDCl_3_) δ 26.0; HRMS (ESI) *m*/*z* calcd for C_14_H_26_N_2_O_3_P [M + H] + 301.1675, found 301.1673.

**2-Cyano-3-(dimethoxy-phosphoryl)-3-phenyl-propionic acid ethyl ester (11).** Compound **11** was obtained according to General Method with 86% yield (268 mg, 0.86 mmol) as colorless semi-solid as a mixture of diastereoismers (ratio 1:2); ^1^H NMR (400 MHz, CHCl_3_) δ 7.48 (dt, *J* = 7.6, 1.9 Hz, 2H), 7.41–7.30 (m, 6H), 4.26 (dd, *J* = 8.7, 6.4 Hz, 1H), 4.16 (q, *J* = 7.1 Hz, 2H), 4.10–4.02 (m, 2H), 3.90–3.80 (m, 1H), 3.79 (d, *J* = 10.9 Hz, 2H), 3.69 (d, J = 11.0 Hz, 3H), 3.66 (d, *J* = 10.7 Hz, 3H), 3.48 (d, *J* = 10.7 Hz, 2H), 1.18 (t, *J* = 7.1 Hz, 3H), 1.07 (t, *J* = 7.1 Hz, 2H); ^13^C NMR (100 MHz, CDCl_3_) δ 164.18, 164.08, 129.24, 128.97, 128.95, 128.83, 128.80, 114.76, 114.66, 63.34, 63.14, 54.15, 54.13, 54.08, 54.06, 53.37, 44.92, 44.15, 43.49, 42.71, 39.38, 39.15, 13.79, 13.63; ^31^P NMR (162 MHz, CDCl_3_) δ 24.45, 24.33; HRMS (ESI) *m*/*z* calcd for C_14_H_19_NO_5_P [M + H] + 312.0995, found 312.0992.

**2-Cyano-3-(dimethoxy-phosphoryl)-3-(4-methoxyphenyl)-propionic acid ethyl ester (12).** Compound **12** was obtained according to General Method with 89% yield (303 mg, 0.89 mmol) as colorless semi-solid as a mixture of diastereoismers (ratio 1:2); ^1^H NMR (400 MHz, CDCl_3_) δ 7.36–7.31 (m, 2H), 7.26–7.21 (m, 1H), 6.80 (ddd, *J* = 8.6, 7.8, 0.7 Hz, 3H), 4.17 (dd, *J* = 8.6, 6.2 Hz, 1H), 4.09 (q, *J* = 7.1 Hz, 2H), 4.03–3.94 (m, 1H), 3.77 (d, *J* = 6.2 Hz, 1H), 3.72 (s, 1H), 3.70 (s, 3H), 3.69 (d, *J* = 1.2 Hz, 3H), 3.61 (s, 3H), 3.58 (d, *J* = 10.7 Hz, 3H), 3.42 (d, *J* = 10.7 Hz, 1H), major steroisomer 1.12 (t, *J* = 7.1 Hz, 3H), minor stereoisomer 1.02 (t, *J* = 7.1 Hz, 1H); ^13^C NMR (100 MHz, CDCl_3_) δ 164.22, 163.98, 130.93, 130.87, 130.47, 130.41, 123.49, 123.43, 122.80, 122.74, 114.81, 114.72, 114.32, 114.30, 114.29, 63.24, 63.04, 54.07, 54.06, 54.00, 53.99, 53.27, 53.20, 43.95, 43.18, 42.51, 41.74, 39.55, 39.29, 39.27, 13.76; ^31^P NMR (162 MHz, CDCl_3_) δ 24.7, 24.6; HRMS (ESI) *m*/*z* calcd for C_15_H_21_NO_6_P [M + H] + 342.1101, found 342.1098.

**Dibenzyl (2-cyanoethyl)phosphonate (13).** Compound **13** was obtained according to General Method with 43% yield (135 mg, 0.43 mmol) as colorless oil; ^1^H NMR (400 MHz, CDCl_3_) δ 7.44–7.29 (m, 10H), 5.12–4.92 (m, 4H), 2.55–2.41 (m, 2H), 2.10–1.95 (m, 2H); ^13^C NMR (100 MHz, CDCl_3_) δ 135.70 (d, *J* = 5.7 Hz), 128.81 (d, *J* = 6.7 Hz), 128.21, 118.25 (d, *J* = 19.1 Hz), 67.95 (d, *J* = 6.6 Hz), 23.38 (d, *J* = 144.0 Hz), 11.42 (d, *J* = 2.8 Hz); ^31^P NMR (162 MHz, CDCl_3_) δ 27.0. NMR data were in accordance with those reported in the literature [79].

**Diethyl (2-cyanoethyl)phosphonate (14).** Compound **14** was obtained according to General Method with 49% yield (94 mg, 0.49 mmol) as colorless oil; ^1^H NMR (400 MHz, CDCl_3_) δ 4.08 (dqd, *J* = 8.1, 7.1, 4.3 Hz, 4H), 2.64–2.52 (m, 2H), 2.07–1.96 (m, 2H), 1.32–1.23 (m, 6H); ^13^C NMR (100 MHz, CDCl_3_) δ 118.37 (d, *J* = 18.1 Hz), 62.34 (d, *J* = 6.7 Hz), 30.81, 22.77 (d, *J* = 144.8 Hz), 16.41 (d, *J* = 5.7 Hz), 11.59 (d, *J* = 3.8 Hz); ^31^P NMR (162 MHz, CDCl3) δ 25.9. NMR data were in accordance with those reported in the literature [79].

**Benzylidenemalononitrile.** ^1^H NMR (400 MHz, CDCl_3_) δ 7.96–7.85 (m, 2H), 7.78 (s, 1H), 7.68–7.58 (m, 1H), 7.52 (dd, *J* = 8.5, 7.0 Hz, 2H); ^13^C NMR (100 MHz, CDCl_3_) δ 160.16, 134.67, 130.99, 130.76, 129.65, 113.83, 112.69, 82.68. NMR data were in accordance with those reported in the literature [80].

## 4. Conclusions

Our research focused on the development of a method for the synthesis of β-phosponate derivatives that would not require the use of cytotoxic and genotoxic reagents. The use of enzymes as catalysts for the reactions to obtain target compounds has been proposed. As a result of the research, it was found that the best biocatalysts are lipases, and among them, lipase from *Candida cylindracea*. New promiscuous activity of lipases in phosphorus–carbon bond formation leading to β-phosphonomalononitriles was presented. Our elaborated protocol provides an efficient, mild, and metal-free synthesis of the target products with a high yield (43–93%). Among the studied derivatives, the compounds 5, 6, 7, and 12 were obtained with the highest yields, which possess halogen atoms or methyl and methoxy groups in aromatic substituent located at beta-position and turned out to be the antimicrobial agents with activity profiles similar to commonly used antibiotics: ciprofloxacin, bleomycin, and cloxacillin (Figure 4 and Figure 9). The results of the presented research are important for understanding the biological properties of the tested β-phosphononitrile derivatives as a function of potential new antibiotics and their toxic effects on Gram-negative bacteria in the face of growing drug-resistance. Our studies also show that the synthesized β-phosphonomalononitriles have lower MIC values compared to well-known antibiotics, which allows us to say that DKPs hold more potential as antibiotic drug candidates due to high anti-bacterial activity for all the tested mutants. The observed results are especially important in the case of the increasing resistance of bacteria to various drugs and antibiotics. Moreover, the obtained compounds constitute a practical platform for further chemical modifications that may favorably translate into their pharmacological as well as pharmacokinetic properties for the development of antibiotics. Additionally, due to the low acquisition costs, they can be an attractive alternative to the currently used antibiotics. 

## Data Availability

On request of those interested.

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
