# Peer review of "Promiscuous Lipase-Catalyzed Knoevenagel–Phospha–Michael Reaction for the Synthesis of Antimicrobial β-Phosphono Malonates"

_ijms, 2022, doi:10.3390/ijms23158819_

Round 1

Reviewer 1 Report

The presented manuscript includes the study of promiscuous lipase-catalyzed Knoevenagel–phospha–Michael reaction for the synthesis of antimicrobial β-phosphono malonates.

The paper is of interest. The results of the work are presented on a good level but the paper contains a huge amount of typos.

1. Please, check the journal requirements for the manuscript structure. Please, insert the 2d section “Materials and methods”. Divide it into reasonable subsections.

2. Existing subsection 2.1 is a part of the Introduction section. Guess this is your 3d section.

3. Paper includes a lot of typos (e.g. lines 197, 226, 228, Fig. 7 and 8 Y-axis, 258, etc.). 

4. Line 205, 236, 239, 244, 272. Please do not use more than 3 refs in one place, otherwise describe the differences.

5. Please use dots for decimal points.

Author Response

Firstly, we would like to express our gratitude to Reviewers for their suggestions that allowed us to considerably improve our manuscript. Please find enclosed for your consideration the revised article ijms-1841688 entitled “Promiscuous Lipase-Catalyzed Knoevenagel–phospha–Michael reaction for the Synthesis of Antimicrobial β-Phosphono Malonates” by Koszelewski et al.

We have revised the text according to the suggestions and we hope that you will now find it suitable for publication in the International Journal of Molecular Sciences. Below, please find the detailed information on the changes in the manuscript with answers to all comments. All changes made in the manuscript were marked up using the “Track Changes” function.

Reviewer 1: The presented manuscript includes the study of promiscuous lipase-catalyzed Knoevenagel–phospha–Michael reaction for the synthesis of antimicrobial β-phosphono malonates.The paper is of interest. The results of the work are presented on a good level but the paper contains a huge amount of typos.

Response: We are grateful for the effort put by the Reviewer to check our work. We have made every effort to eliminate language errors and typos.

Reviewer 1: 1. Please, check the journal requirements for the manuscript structure. Please, insert the 2d section “Materials and methods”. Divide it into reasonable subsections.

Response: We are very grateful for this remark. As suggested by the Reviewer, the layout of the work has been corrected in line with the publishing house's recommendations. There is a section entitled "Materials and methods" in the manuscript.

Reviewer 1: 2. Existing subsection 2.1 is a part of the Introduction section. Guess this is your 3d section.

Response: We are very grateful for this remark. As suggested by the Reviewer, the subsection 2.1 was revised and modified. Some part of this section was moved in to the Introduction section.

Reviewer 1: 3. Paper includes a lot of typos (e.g. lines 197, 226, 228, Fig. 7 and 8 Y-axis, 258, etc.).

Response: We are grateful for the effort put by the Reviewer to check our work. We have made every effort to eliminate language errors and typos.

Reviewer 1: 4. Line 205, 236, 239, 244, 272. Please do not use more than 3 refs in one place, otherwise describe the differences.

Response: We are very grateful for this remark. As suggested by the Reviewer, the number of refs in one place was reduced and fulfils the publishing house's recommendations.

Reviewer 1: 5. Please use dots for decimal points.

Response: We are very grateful for this remark. Due to problems with the regionalization of the operating system used to prepare the charts, we were unable to use periods instead of commas. We ask the publisher for help in solving this problem

Reviewer 2 Report

Reviewers' comments:

Manuscript ID: ijms-1841688

Title: Promiscuous Lipase-Catalyzed Knoevenagel–phospha–Michael reaction for the Synthesis of Antimicrobial β-Phosphono Malonates.

Manuscript Type: Article.

Reviewers' comments:

The manuscript describes the Promiscuous Lipase-Catalyzed Knoevenagel–phospha–Michael reaction for the Synthesis of Antimicrobial β-Phosphono Malonates. The manuscript needs a detailed editing. Some markings are made to just illustrate the extent of editing needed.

The authors need to consider the following comments

- Author should check sectional numbers throughout the manuscript.

- In the Abstract: the authors need to improve with more specific short results and conclusions, i.e. academic novelty or technical advantages.

- Keywords: add more suitable keywords.

- The introduction - authors should elaborate their introduction section by citing few more relevant references.

- 2.3. Analysis of R2–R4 E. coli strains modified with β-phosphonate derivatives - section should be detailed.

- 3.4. R2-R4 E. coli strains with tested β-phosphonate derivatives - section should be detailed.

- 3.1. Microorganisms and Media – section should be detailed.

- 5. Conclusions…to….4. Conclusions

- Conclusions, the author should add some qualitative data of the results and should rebuild to let it fluent.

- Make all references in same format for volume number, page number and journal name, because it is difficult to searching and reading.

- Minor English corrections is required throughout the manuscript.

Based on these, I advise the authors to rectify the above mentioned errors and we hope to re-evaluate the revised manuscript.

Author Response

Firstly, we would like to express our gratitude to Reviewers for their suggestions that allowed us to considerably improve our manuscript. Please find enclosed for your consideration the revised article ijms-1841688 entitled “Promiscuous Lipase-Catalyzed Knoevenagel–phospha–Michael reaction for the Synthesis of Antimicrobial β-Phosphono Malonates” by Koszelewski et al.

We have revised the text according to the suggestions and we hope that you will now find it suitable for publication in the International Journal of Molecular Sciences. Below, please find the detailed information on the changes in the manuscript with answers to all comments. All changes made in the manuscript were marked up using the “Track Changes” function.

Reviewer 2: The manuscript describes the Promiscuous Lipase-Catalyzed Knoevenagel–phospha–Michael reaction for the Synthesis of Antimicrobial β-Phosphono Malonates. The manuscript needs a detailed editing. Some markings are made to just illustrate the extent of editing needed. The authors need to consider the following comments

Reviewer 2: - Author should check sectional numbers throughout the manuscript.

Response: We are very grateful for this remark. Sectional numbers were revised and corrected.

Reviewer 2: - In the Abstract: the authors need to improve with more specific short results and conclusions, i.e. academic novelty or technical advantages.

Response: We are very grateful for this remark. Due to the Reviewer suggestion the Abstract was modified.

Reviewer 2: - - Keywords: add more suitable keywords.

Response: We are very grateful for this remark. Due to the Reviewer suggestion few more suitable keywords were provided.

Reviewer 2: - The introduction - authors should elaborate their introduction section by citing few more relevant references.

Response: We are very grateful for this remark. Due to the Reviewer suggestion the Introduction part was revised and modified.

Reviewer 2: - 2.3. Analysis of R2–R4 E. coli strains modified with β-phosphonate derivatives - section should be detailed.

The description has been supplemented with appropriate citations and descriptions in the manuscript

Reviewer 2: - 3.4. R2-R4 E. coli strains with tested β-phosphonate derivatives - section should be detailed.

The description has been supplemented with appropriate citations and descriptions in the manuscript

Reviewer 2: - 3.1. Microorganisms and Media – section should be detailed.

Response: We are very grateful for this remark. Due to the Reviewer suggestion more data regarding microorganism and media was provided.

Reviewer 2:5. Conclusions…to….4. Conclusions

Response: We are very grateful for this remark. It was corrected.

Reviewer 2:Conclusions, the author should add some qualitative data of the results and should rebuild to let it fluent.

Response: We are very grateful for this remark. Due to the Reviewer suggestion Conclusions were revised and modified.

Reviewer 2: - Make all references in same format for volume number, page number and journal name, because it is difficult to searching and reading.

Response: We are very grateful for this remark. It was corrected. References have been formatted as per the publisher's requirements

Reviewer 2:Minor English corrections is required throughout the manuscript.

Response: We are very grateful for this remark.  We have made every effort to eliminate linguistic errors.

Round 2

Reviewer 2 Report

The suggested changes and improvements have been done.